# Co-design of "Baatcheet," a peer-supported, web-based storytelling intervention for young people with common mental health problems in India

Pattie P. Gonsalves[1] ⬤, Dhriti Mittal[1], Shruti Aluria[1], Aarushi Khan[1], Eshita Razdan[1], Priyambada Kashyap[1], Navvya Rahate[1], Manek D'Silva[2], Sonaksha Iyengar[3], Faith Gonsalves[4], Sweta Pal[1], Salik Ansari[1], Clio Berry[5] and Daniel Michelson[6,7]

[1]Youth Mental Health Group, Sangath, New Delhi, India; [2]Wicked & Wise Entertainment, Bangalore, India; [3]Sonaksha, Bangalore, India; [4]Department of Global Health and Social Medicine, Harvard Medical School, Massachusetts, USA; [5]Brighton and Sussex Medical School, UK; [6]Department of Child and Adolescent Psychiatry, Institute of Psychiatry, Psychology and Neuroscience, King's College London, London, UK and [7]NIHR Maudsley Biomedical Research Centre, South London and Maudsley NHS Foundation Trust and King's College London, London, UK

## Research Article

**Keywords:**
youth; storytelling; digital intervention; peer support; anxiety; depression

**Corresponding author:**
Pattie P. Gonsalves;
Email: pattie.1.gonsalves@kcl.ac.uk

## Abstract

**Background:** Engaging with personal mental health stories has the potential to help people with mental health difficulties by normalizing distressing experiences, imparting coping strategies and building hope. However, evidence-based mental health storytelling platforms are scarce, especially for young people in low-resource settings.
**Objective:** This paper presents an account of the co-design of 'Baatcheet' ('conversation' in Hindi), a peer-supported, web-based storytelling intervention aimed at 16–24-year-olds with depression and anxiety in New Delhi, India.
**Methods:** Development comprised three stages: (1) establishing a logic model through consultations with a Young People's Advisory Group ($N = 11$) and a stakeholder reference group ($N = 20$); (2) elaborating intervention guiding principles and components through focus group discussions and co-design workshops ($N = 42$); and (3) user-testing of prototypes.
**Results:** The developmental process identified key stakeholder preferences for an online, youth-focused mental health storytelling intervention. Baatcheet uses an interactive storytelling website containing a repository of personal stories about young people's experiences of depression and anxiety. This is offered alongside brief support from a peer.
**Conclusions:** There are few story-based interventions addressing depression and anxiety for young people, especially in low-resource settings. Baatcheet has the potential to deliver engaging, accessible and timely mental health support to young people. A pilot evaluation is underway.

## Impact statement

This study describes the co-design of Baatcheet, a peer-supported, web-based mental health storytelling intervention for 16–24-year-olds in India. The intervention was conceptualised to contain a collection of first-person mental health stories from young people about living with, managing and/or recovering from anxiety and depression presented alongside tools for users to search for, reflect on and apply learning from relevant stories, and brief contacts with a peer supporter. The intended outcomes include improvement of mental health outcomes and the use of adaptive coping skills with ultimate impacts aimed at reducing the high burden on the public mental health system in contexts like India. This study is the first systematic attempt to co-design a digital mental health storytelling intervention with young people. Such interventions hold promise in India and other low-resource contexts where there is an urgent need to provide non-stigmatising and resource-efficient interventions to improve young people's mental health. A subsequent study is assessing the acceptability, feasibility, and potential impacts of Baatcheet in a target population of 16–24-year-olds in schools, universities and community centres.

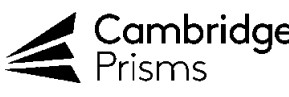


## Introduction

### Background

Drawing on evidence for social contact as a mechanism to reduce stigma (Corrigan, 2012; Corrigan et al., 2016; Goodwin et al., 2021), mental health stories (i.e. first-person narratives describing events or actions related to lived experiences of mental health problems) can be a compelling and effective way to improve a range of mental health outcomes (Llewellyn-Beardsley et al., 2019; McGorry et al., 2024; Reinke et al., 2004). Impacts have most commonly been studied

for storytellers/narrators and include reduced self-stigma (Corrigan, 2012; Corrigan et al., 2016; Goodwin et al., 2021; Nurser et al., 2018), improved quality of life (Corrigan and Shapiro, 2010; Slade et al., 2024), personal empowerment (Corrigan and Shapiro, 2010), enhanced social support (Bos et al., 2009), increased hopefulness (Shaw and Homewood, 2015), and reduced mental health symptoms (Hundert et al., 2021; Kahn and Garrison, 2009; Kahn and Hessling, 2001; Niederkrotenthaler and Till, 2020; Ofoegbu et al., 2021). Mental health stories are also routinely deployed in health staff training, for example, to influence skills and attitudes that are valued by service users (Chambers et al., 2013), and in a variety of mental health stigma reduction programmes (Alvarado-Torres et al., 2023; Makhmud et al., 2022).

Delivery of mental health stories via digital platforms has been identified as a particularly useful strategy especially for individuals with limited access to peers and for reducing social isolation (Rennick-Egglestone et al., 2019). The increasing public availability of digital mental health stories offers an important opportunity to provide support to people through a new form of mental health intervention especially in the context of global challenges such as long waiting lists and limited in-person mental health care options, particularly in low resource settings (Rennick-Egglestone et al., 2020). Despite the widespread use of mental health stories, there is very little research evaluating the effects of mental health stories on recipients or readers who are experiencing mental health difficulties of their own (Rennick-Egglestone et al., 2019) although some studies suggest that storytelling interventions demonstrate strong face validity for improving anxiety and depression outcomes for recipients (Gonsalves et al., 2023).

Most recently, the largest single evaluation of a mental health storytelling platform ('Narrative Experiences Online'; NEON) was conducted in the UK with adults (Slade et al., 2024). NEON tested whether having online access to people's real-life stories of recovery from mental ill health could be helpful for people experiencing psychosis or other mental health problems, and those who offer them care. NEON was designed through synthesis of previous studies, consultation with living/lived experience advisory groups and acceptability and feasibility testing studies. The intervention consists of using a website comprising over 600 previously published stories (e.g. in pamphlets, books or online or donated to the study) from people who have experienced mental health problems. A randomised controlled trial ($N$ = 1,023) found that access to NEON significantly increased quality of life and the presence of meaning in life for participants relative to a usual care condition. Findings also showed that NEON was feasible to independently use by participants and even encouraged the sharing of personal narratives by several participants themselves (Slade et al., 2024). Comparable mental health storytelling platforms have not been specifically designed or adapted for young people, notwithstanding that many young people use social media and other informal channels to share their own mental health stories and to seek out engagement with others. Further, there is scarce research on youth mental health storytelling in low-and middle-income countries (LMICs) (Gonsalves et al., 2023), despite a long tradition of using culturally sensitive narrative approaches to address other public health priorities (Petraglia, 2007).

### The 'Baatcheet' intervention

The current paper reports on the participatory development of 'Baatcheet' (meaning 'conversation' in Hindi). Baatcheet is a web-based mental health storytelling intervention that aims to provide engaging, accessible and timely mental health information and support to young people in India, a lower middle-income country that comprises 20% of the global population of young people. Mental health needs greatly outstrip existing service capacity and mental health stigma further limits access to care, resulting in a treatment gap of >90% (Singh et al., 2023).

Baatcheet has its origins in an archive of stories collected via a national public engagement campaign, 'It's Ok To Talk' (Gonsalves et al., 2019; Sangath, 2024). This initiative was launched in 2017 by the community mental health organisation, Sangath, to promote awareness of, and advocacy for, youth mental health issues. A corresponding website (www.itsoktotalk.in) was produced collaboratively with young people aged 15–24 years, including a section that solicited and published a collection of first-person stories from young people with living/lived experience of mental health problems. In total, over 175 stories were published between 2017 and 2023. Authors were given the choice to submit any kind of media and publish anonymously or to include their name, location and gender. A content analysis of the stories identified prominent themes of hope and acceptance, loneliness, isolation and a strong desire for connection (Gonsalves et al., 2019), echoing common content domains of youth mental health stories obtained in high-income countries (Llewellyn-Beardsley et al., 2019). Submissions additionally touched on sociopolitical determinants of psychological distress, offering a richly contextualised understanding of mental health risks and protective factors. Although not formally evaluated, the story section of www.itsokto talk.in achieved high levels of user engagement based on routinely collected web analytics (Sangath, 2017, 2020), leaving open the possibility that such narratives could be the basis for a focused intervention.

Baatcheet was thus conceived as a web-based intervention incorporating a personalisable library of mental health stories, through which participants could access personally relevant content. It was developed in the Indian context, where mental health problems, particularly depression and anxiety, are the leading mental health concerns for young people (Gururaj G., 2016; UNICEF, 2021). Initial modelling by the development team recognised the importance of brief, synchronous guidance, following well-established findings about poor engagement with purely self-directed digital interventions as compared with guided interventions offering digital and interpersonal modalities (Garrido et al., 2019; Grist et al., 2018; Halsall et al., 2023; Hollis et al., 2017; Miyamoto and Sono, 2012). In line with research that supports the role of peers as an accessible and easy-to-use approach in addressing psychological issues in young people (Ng, 2012; Pointon-Haas et al., 2023), brief peer guidance was identified as the optimal way of offering this guidance.

The goal of the current study was to co-design a detailed intervention specification that drew from these initial concepts. We applied guidelines for mental health intervention co-design which emphasise active collaboration between researchers and end users to result in interventions that are more engaging, satisfying and effective in real-world settings (Bear et al., 2022; O'Cathain et al., 2019; Thabrew et al., 2018). Our specific objectives were to:

1. understand the preferences of young people and other stakeholders regarding the narrative content, appearance, functionality and supportive guidance for a web-based mental health storytelling intervention intended to address depression and anxiety among 16–24-year-olds in India; and

2. apply this understanding to undertake iterative development and user testing of the intervention.

## Methods

### Design

The process of developing the Baatcheet intervention was conducted in three interconnected stages over one year (January–December 2023) (see Figure 1). The stages comprised: (1) establishing an intervention logic model based on consultations with a Young People's Advisory Group (YPAG) and a stakeholder reference group; (2) elaborating intervention guiding principles and more detailed specifications through focus group discussions (FGDs) and iterative co-design workshops with prospective end-users; and (3) user-testing and refinement of prototypes. We followed guidelines for the development of person-centred digital mental health interventions and complex health interventions (Mohr et al., 2017; Thabrew et al., 2018; Yardley et al., 2015), which emphasise the importance of user input at each stage (Fleming et al., 2019; Prebeg et al., 2023; Wight et al., 2016).

### Setting

Research activities were conducted at five sites in New Delhi: two secondary schools serving low-income communities, one public medical university and two community centres in urban villages managed by a community-based organisation (CBO). These sites were selected for the current study to ensure coverage across a variety of youth settings.

### Involvement of youth advisors

Consistent with guidance about the involvement of young people as 'co-actors' in health research (Sellars et al., 2021; Vojtila et al., 2021), we recruited a Young People's Advisory Group (YPAG) comprising 11 young people (9 female, 2 male) aged 16–25 years.

Youth Advisors contributed to all three stages of the design process. Their specific contributions included participation in the overall planning and conduct of the research, as opposed to taking on the role of research 'participants,' for example, they did not supply 'data' but advised on co-design activity plans and peer support plans and provided inputs on data analysis.

The YPAG selection process involved an open call on social media, online newsletters and emails to local youth organisations. Interested individuals completed an application form, followed by an interview with shortlisted candidates to assess motivation, interest in mental health research and ability to participate in monthly meetings and workshops. Applications were screened by a panel comprising one study team member and an external independent reviewer. Selection was focused on the recruitment of a diverse group, including those with lived/living experience of mental health problems and those belonging to socially disadvantaged groups.

The YPAG participated in six sessions over a year (see Supplementary Material-Annexure 1), with additional contributions via email and WhatsApp. YPAG sessions were conducted at the host institution's office at mutually agreed-upon times and adhered to mutually agreed principles for group participation. We included an orientation and a brief training on the role of mental health stories. Subsequent sessions included interactive group activities to review and update the logic model, refine intervention goals, review and comment on co-design workshop plans, review features for the website and peer delivery and the final selection of included stories. Co-design findings were presented to the YPAG, who provided feedback and final sign-off on design decisions. To maximise comfortable and safe participation, regular breaks and refreshments were included in all sessions. Information about free mental health support services (e.g. tele counselling helplines) and debriefing sessions was also provided as needed. YPAG members were provided with honorarium payments, recognition on the Baatcheet website and invitations to external presentations on the project. Three Youth Advisors volunteered their time as co-authors of this paper on behalf of the wider YPAG.

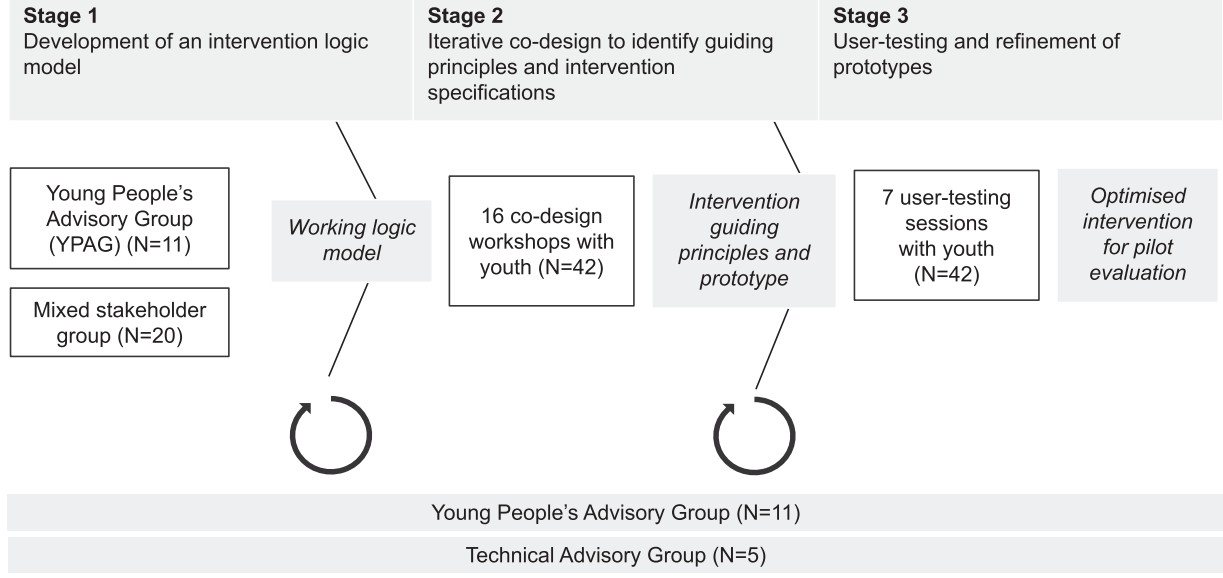

**Figure 1.** Overview of the intervention development process.

### Involvement of a technical advisory group (TAG)

A five-member Technical Advisory Group (TAG) consisting of experts in qualitative research, narrative interventions, bioethics and lived experience was involved in reviewing research plans and iterations of the intervention prototype. Their inputs were focused on refining co-design strategies, assessing potential risks and ensuring the intervention aligned with best practices and the needs of the target population (see Figure 1). Four members of the TAG volunteered their time as co-authors of this paper on behalf of the wider TAG.

### Stage 1: Development of a logic model

#### Participants

A group of 20 mixed stakeholders (four student volunteers, three narrative therapists, three psychologists, two government representatives, two non-governmental organisation (NGO) representatives, one school counsellor and one university professor) and four members of the Technical Advisory Group (TAG) (FG, SP, SA and HS) participated in a day-long consultation workshop along with the research team (see Supplementary Material-Annexure 2). Contributors were identified through the institutional networks of the host organisation (Sangath) and collaborating study sites.

#### Procedures

This consultation obtained feedback on an initial intervention outline prepared by the study team and YPAG. Attendees worked in smaller groups of 5–6 individuals using the 'MoSCoW' (Must have, Should have, Could have and Won't have) prioritisation tool (Clegg and Barker, 1994) to respond to 'How might we' style questions related to promoting intervention safety, uptake, motivation and participation challenges. For example, '*How might we reduce the access barriers that young people may face in participating in the Baatcheet programme*'. Detailed process notes were completed in real time.

#### Analysis

Feedback from the consultation was summarised into predetermined categories (related to intervention inputs, activities and outcomes) and represented diagrammatically as a logic model (see Figure 2 and Supplementary Information).

### Stage 2: Intervention co-design

#### Participants

A cohort of 42 young people (21 male, 21 female; aged 16–22 years) took part in focus group discussions (FGDs) and iterative co-design activities (see Table 1), which respectively aimed at formulating intervention guiding principles followed by a more detailed intervention 'blueprint'. Participants were recruited from the collaborating sites in response to printed posters, digital distribution of the same poster via WhatsApp groups and brief in-person announcements by the study team. To capture a range of perspectives, we tried to recruit a balanced sample with regard to age, gender and site. No specific eligibility criteria were used to select participants, except for language proficiency in English and Hindi and an interest in the topic area. Students with living/lived experience and belonging to socially disadvantaged groups were encouraged to apply.

#### Procedures

(i) FGDs. One FGD was conducted at each site (see Supplementary Material-Annexure 3). Discussions were focused on understanding awareness and prior experiences of online platforms where mental health stories are shared, favoured narrative styles and content areas, and preferred features and functions of the prospective Baatcheet website and associated support. Each FGD lasted 60–90 minutes and was conducted jointly by two study team members (PPG and DMittal).

(ii) Co-design workshops. Each participant took part in two co-design workshops (see Supplementary Material-Annexure 4).

| Inputs | Outputs | | Outcomes | | |
| --- | --- | --- | --- | --- | --- |
| | Activities | Participants | Short-term | Intermediate | Long-term |
| Co-designed digital intervention aimed at addressing depression and anxiety among youth aged 16-24 years.<br><br>Brief guidance from a peer supporter. | First-person mental health stories about living with, managing and/or recovering from anxiety and depression.<br><br>Tools for users to search for, reflect on and apply learning from relevant stories.<br><br>Resources to find external support.<br><br>Contacts with a peer supporter. | Young people aged 16-24 years experiencing mental health problems. | Increased problem normalization.<br><br>Increased reflective capacity.<br><br>Increased social connectedness.<br><br>Increased hopefulness. | Emotional: Improvement in mental health outcomes (depression, anxiety and social functioning).<br><br>Cognitive: Re-appraisal of stressors and coping resources.<br><br>Behaviour change: Use of adaptive coping skills. | Intervention available as an open-access resource and integrated into school or university mental health delivery.<br><br>Reduced burden on mental health services via peer delivery model. |

**Figure 2.** Logic model.

**Table 1.** Co-design participant characteristics

| Research site | Participants (*N*) | Number of sessions* | Age range (years) | Female: Male (*N*) |
|---|---|---|---|---|
| University Group 1 | 8 | 5 | 18–22 | 1F:7M |
| University Group 2 | 8 | 4 | 18–22 | 4F:4M |
| School 1 | 10 | 4 | 16–18 | 7F:3M |
| School 2 | 7 | 5 | 16–18 | 2F:5M |
| Community-based organisation (CBO) | 9 | 5 | 17–20 | 7F:2M |
| Total | 42 | 23 | 16–22 years | 21F:21M |

*Sessions consisted of one focus group discussion, two co-design workshops and up to two user testing sessions.

In the first workshop, participants worked in small groups to design a paper prototype of the Baatcheet website. They addressed questions related to the presentation, categorisation and filtering of stories, as well as identifying strategies to enhance the interactivity of the website. The second workshop focused on peer support. Participants worked on case vignettes to identify and prioritise peer supporter characteristics and role specifications. Examples of questions included: 'How can the Baatcheet peer supporter build a safe and trusting relationship with an intervention user in the first meeting?' and 'What information should the Baatcheet peer supporter highlight to an intervention user about the programme?' Each workshop lasted 90–120 minutes and was conducted jointly by two researchers (DMittal, SA and/or AK).

### Analysis

FGDs and workshops were audio-recorded, transcribed and summarised in contemporaneous process notes. Transcripts and process summaries were analysed and triangulated around the research objectives using an integrated inductive-deductive approach to content analysis and through thematic and mapping techniques (Bradley 2007). In the first instance, coders (PPG, DMittal, AK and SA) familiarised themselves with the data by reading and rereading transcripts and summaries to identify provisional codes and higher-level themes/sub-themes, assisted by Nvivo Version 12 software. Initial codes/themes were revised and consolidated through group discussion.

### Stage 3 – Prototype user-testing and refinement

#### Participants

The same participants from Stage 2 were invited to take part in user-testing of 'clickable' prototypes which simulated the planned intervention website.

#### Procedures

Each group took part in up to two user-testing sessions depending on their availability to attend sessions (see Table 1). Participants who took part in two sessions were shown sequentially updated versions of the website prototype. Feedback was obtained through 'think aloud' discussions (Eccles and Arsal, 2017) and the completion of the System Usability Scale (SUS) (Brooke, 1995). The SUS comprises 10 items on a 5-point Likert scale ranging from 1 (strongly disagree) to 5 (strongly agree), with half of the items reverse-scored. For administration in this study, the scale was adapted by replacing the term 'system' with 'website' in each item. Each session lasted for around 1 hour and was conducted jointly by two researchers (DMittal, SA and/or AK).

#### Analysis

Prototype feedback was summarised around key website features for each iteration of the website (see Table 4).

### Results

#### Stage 1: Development of a logic model

Stakeholder feedback was utilised to prepare a preliminary logic model (see Figure 2) which suggests short-term and intermediate impacts on problem normalisation, reflective capacities, social connectedness and mental health symptoms and ultimate impacts on the public mental health system.

#### Stage 2: Intervention co-design

Thematic analysis of FGDs identified six high-level guiding principles for the intervention (see Figure 3). More detailed specifications were developed through co-design workshops, from which findings were organised around four themes related to: (1) experiences and impacts of mental health storytelling; (2) story contents; (3) preferences for website design; and (4) preferences for peer support. A summary of

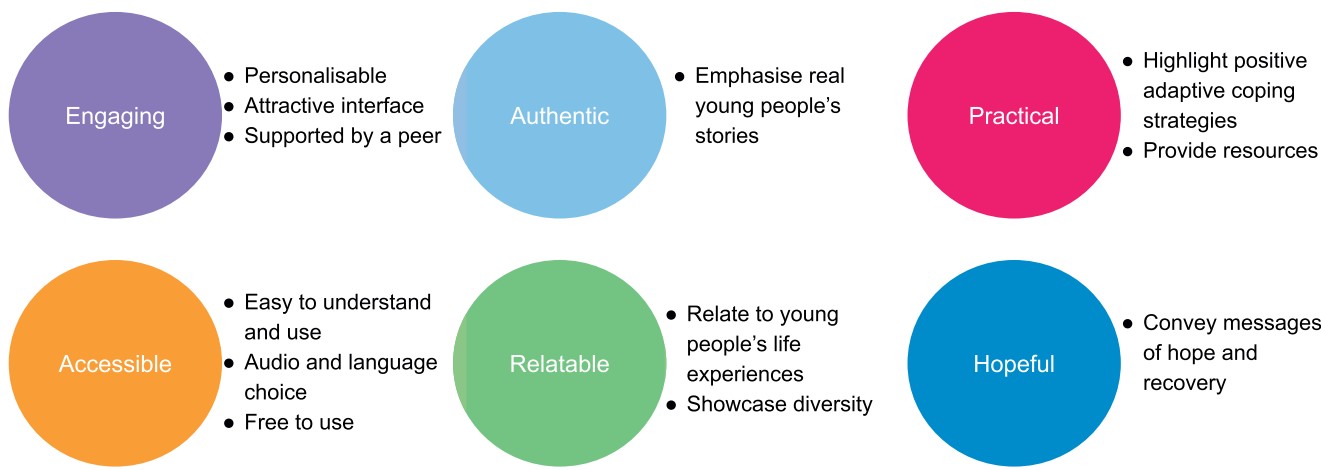

**Figure 3.** Intervention guiding principles.

**Table 2.** Co-design workshop themes and design implications

| | Theme and sub-theme | Examples of participant quotes | Implications for website design |
|---|---|---|---|
| 1 | Experiences and impacts of mental health storytelling | *I want to share my own story. About 2–3 years ago my parents got divorced. I got a bit depressed and just could not move on. I felt really stressed and did not want to talk to anyone. (School student, 17 years)*<br>*Stories can help you get things off your chest and make you feel lighter. If you can put out some content or your story, you might also feel like you are helping others and boosting your own self-confidence (College student, 19 years)*<br>*I reached out* via *Reddit to groups who talked about problems like depression and ADHD…I gained a lot of friends through this. (College student, 19 years)*<br>*Everyone's story is unique…when we know that they are going through a similar experience, it's always comforting. You know someone else is experiencing that, and it's not "your" problem, but it's quite general and this is so comforting to know. (College student, 19 years)* | • Highlight the benefits of mental health storytelling on the home page. |
| 2 | Story contents | *I would prefer if someone shared their practical tools and experiences like methods to come out of depression and not just motivational talking. (College student, 19 years)*<br>*I would like to see stories in a "how to" style like how to recover from study problems, family problems, social anxiety, a broken heart, and exam-related problems. (School student, 17 years)*<br>*I would be open to listening to anyone's experiences because you do not know what you will get to learn. (College student, 20 years)* | • Present 'categories' or 'collections' of stories grouped using a 'How to…' framing (for example, 'How to overcome depression' or 'How to recover from a broken heart') to help users find stories of their interest more quickly. |
| 3 | Preferences for website design | | |
| a | Tone and style | *Use pictures to represent stories or themes. Visual cues would make it easier for users to navigate and select their preferred content, as images can convey information more intuitively." (School student, 18 years)*<br>*Stories could be enriched with multimedia elements like audio, video, animations, and even comics to provide a more immersive experience. (School student, 17 years)*<br>*Use basic understandable language so you do not need to Google any of the terms. (College student, 21 years)* | • Include relevant illustrations wherever appropriate.<br>• Include an option to select from different multimedia story formats (e.g. text or video).<br>• Include a title and brief description for each story.<br>• Include basic information (e.g. age, gender and location) for each story narrator.<br>• Ensure simple language. |
| b | Functionality | *I think the platform should be designed in a way that encourages users to delve deeper into challenging topics and engage meaningfully with the content. (College student, 20 years)*<br>*It should be available offline and compatible with low bandwidths. (School student, 17 years)*<br>*The navigation within the website should be simple. Have an easy login without too many steps to enter the website. (School student, 17 years)*<br>*Reading others' stories may make you feel like sharing but it needs to be moderated. (College student, 19 years)* | • Allow users to reflect on the content (e.g. via reflective questions).<br>• Design the website to work well on low bandwidths.<br>• Enable active engagement with stories (e.g. via a reaction button).<br>• Include an option for a user to share their own story if they wish. |
| c | Personalisation | *Include language as a category for users who prefer to access stories in their native language. (School student, 18 years)*<br>*Stories can be organised by themes, allowing users to click on their specific problems and find related stories. (CBO participant, 20 years)*<br>*Trigger warnings must be included to alert users about sensitive or distressing content in certain stories. (CBO participant, 21 years)* | • Include an option for a user to select a preferred language.<br>• Include an option to filter based on narrator and story characteristics.<br>• Include content and trigger warnings and the ability to filter and hide stories with triggers. |
| d | Trust and safety | *The website should not have any advertisements. (School student, 17 years)*<br>*Include information about mental health problems like depression and anxiety. (School student, 17 years)*<br>*Telling your story should give you the choice to be anonymous. (School student, 18 years)* | • Include an easy-to-understand data and privacy policy.<br>• Provide information about self-help resources and helplines.<br>• Include an option to delete personal or saved information.<br>• Include an option to share stories anonymously. |

co-design workshop themes and design implications is presented in Table 2.

### (1) Experiences and impacts of mental health storytelling

Experiences of online mental health stories were limited to reading news reports of high-profile celebrity suicides and browsing or contributing to anonymised postings to community platforms such as Reddit. Some participants also noted offline situations in which peers and family members had spontaneously described experiences of mental health difficulties arising from stressful life events. These 'stories' included accounts of failing competitive national exams, sudden unemployment and difficult family relationships.

For narrators (i.e. someone who shares their story), participants highlighted storytelling as a potential strategy to 'feel lighter', 'feel heard', and to be supported or cared for by others. However, participants recommended careful consideration of when to disclose, and to whom. As recipients (i.e. someone who reads or hears others' mental health stories), participants highlighted multiple benefits such as learning coping strategies from others; building empathy for those with unfamiliar experiences; feeling less alone through finding community; feeling inspired and being motivated to share one's own story.

### (2) Story contents

Participants highlighted five key characteristics of mental health stories for the proposed Baatcheet intervention: (i) authentic, first-person narratives rather than hypothetical or third-person accounts; (ii) recovery-oriented, hopeful messages that change is possible and a better future can be achieved; (iii) at the same time, a sense of realism that not all problems are tractable and life can be rewarding despite the challenges posed by mental ill-health; (iv) contextualised descriptions of practical and actionable coping strategies; and (v) a focus on relatable problems and contexts that resonate with users from diverse backgrounds. Story characteristics identified were used to develop a checklist to characterise stories for inclusion in the intervention website (see Supplementary Information).

### (3) Preferences for website design

(i) Tone and style. There was consensus about using a vibrant and colourful visual style, with text accompanied by suitable imagery. Participants recommended catchy and simple story titles, the inclusion of story summaries, easy-to-read formatting, and an absence of jargon about mental health problems 'so you do not need to Google any of the terms.' In terms of media choices, participants recommended having a range of formats (e.g. audio, video, text, art or animation) to cater to different user preferences.

(ii) functionality. There was a unanimous preference for including interactive features, particularly those that added to a feeling of 'connecting to people who are going through similar challenges and creating a space where they can find solace and support from others who understand their experiences.' As one example, Emojis and 'reaction' buttons were suggested as a means of signalling engagement with stories and potentially offering encouragement to story narrators. Participants, however, recommended presenting these without tallies to prevent competitiveness and pressure to get "likes." Providing audio versions for written stories and subtitles for videos was recommended to enhance accessibility. Participants agreed that the website should be free to use and compatible with low internet bandwidth and low-end smartphones. They emphasised the importance of 'easy login without too many steps to enter the website.' There was consensus on not offering a public commenting feature for stories to protect story narrators from any negative comments.

(iii) Personalisation. Participants wanted the website to feel as though it speaks to users' individual needs and problems. Suggested tools/features for personalisation included detailed filters for story content (e.g. by problem type or story theme) and narrator characteristics (e.g. age, gender, location, sexual identity), as well as adjustable settings for content warnings. On the other hand, some participants expressed openness and curiosity about stories from narrators of all backgrounds. Other recommendations concerned the use of bi-lingual content in English and Hindi.

(iv) Trust and safety. Participants recommended that the website should exist independently of social media platforms like Instagram or YouTube, reflecting general mistrust of social media platforms and 'influencers' who publish personal content as part of paid endorsements. Relatedly, participants also expressed strong views against having any kind of advertisements on the website. To counter any mistrust, they recommended the inclusion of detailed information about the project team and their affiliated institutions on the home page.

Participants raised additional concerns related to data security, confidentiality and user welfare. They recommended soliciting minimal personal data at the time of account creation and obtaining explicit consent from users about data use, with all data use and privacy policies published in lay terms. Offering anonymity in sharing one's own story on the website was seen as an important factor in gaining the trust of prospective contributors. Other suggestions related to the personal welfare of users through automated reminders to take breaks while reading stories and signposting to external mental health support services.

### (4) Preferences for peer support

(i) Role of peer supporters. Participants strongly favoured a 'human element' to facilitate the use of the intervention website. Discussion centred on a peer supporter role ('a buddy and not a counsellor') offering relational, motivational and technical support to website users. Participants saw this role as encompassing social connection, advocacy for the intended benefits of the website and technical know-how about how to use the website, for example, offering advice on how to create an account, how to locate the different website features, how to find the most relevant stories, and how to share one's own story. Participants also outlined benefits that the peer supporters could gain through participation, such as enhanced mental health knowledge and new skills related to communication and active listening. Imagined challenges for peer supporters included expectation management ('peer supporters may not have all-encompassing answers'), power imbalances between peer supporters and intervention users, difficulty juggling personal schedules/other commitments alongside a peer supporter role, self-care, and access to appropriate supervision.

(ii) Peer supporter characteristics. Participants outlined preferences for peer supporters who had lived/living experiences of mental health difficulties to help build relatability and comfort of prospective intervention users. Participants expressed preferences for peer supporters who were slightly older (by one or two years) and cautioned against a selection of peer supporters from the same setting, such as classmates. They strongly endorsed qualities such as warmth and approachability in helping 'to create a safe and respectful environment' as well as specific skills such as active listening and cultural sensitivity.

(iii) Format and frequency of support. Participants recommended weekly meetings with a peer supporter for up to one month, with intervention users having the option to contact peer supporters directly if they needed additional support. They felt this duration would fit within existing school or university schedules. Individual meetings were favoured over group meetings to promote confidentiality and provide a more personalised experience. There were varied preferences for mode of contact (e.g. in-person,

virtually/telephonically, text messaging), with the presumption that users should be able to decide for themselves.

### Stage 3 – Prototype user-testing and refinement

An initial website prototype was produced using insights from Stage 2. The features are summarised in Table 3. Participants involved in user-testing acknowledged the user-friendly nature of the website, the attractive and simple interface, straightforward navigation and story presentation, and the convenience of having audio files accompany stories (see Figure 4). They also appreciated the option to react to and save specific sections of stories. Several potential improvements were also identified and incorporated into a second prototype (see Table 4). Participants suggested less abstract imagery and less text. They also provided additional information about the peer support element (which was not part of user testing). The mean System Usability Score (SUS) of the updated prototype was 76.5, indicating above-average usability.

### Discussion

This paper outlines the co-design of Baatcheet, a peer-supported, web-based mental health storytelling intervention for 16–24-year-olds in India. According to our provisional logic model, the intervention was conceptualised to contain a collection of first-person mental health stories from young people about living with, managing and/or recovering from anxiety and depression presented alongside tools for users to search for, reflect on and apply learning from relevant stories, and brief contacts with a peer supporter. The intended outcomes include improvement of mental health outcomes and the use of adaptive coping skills with ultimate impacts aimed at reducing the high burden on the public mental health system in contexts like India.

Engagement, authenticity, offering practical information, accessibility, relatability and hopefulness emerged as guiding principles that were translated into a working prototype which was refined through user-testing activities. Participants wanted credible, culturally and contextually congruent stories that would be optimally relevant to their particular mental health problems and social circumstances. The ability to filter stories and the inclusion of reflective tools were recommended to help users apply story insights to their own lives. This is aligned with findings from other health-related storytelling studies where reflection has been cited as a key activity in creating cognitive and affective associations that influence desired behaviours (Ng et al., 2024; Slade et al., 2021). The Baatcheet intervention however differs from previous interventions

**Table 3.** Overview of Baatcheet intervention prototype

| Website features | Description |
|---|---|
| Home and about | A 'Home' page containing information about the intervention, how to use it and potential benefits of participation. The 'About' section includes testimonials from the development team. |
| Login | A login page provides the option to sign up using a mobile number or email address. |
| Story library | After logging in, users are taken directly to the 'Story Library' page which is one of the key features of the website. This page contains story suggestions based on the user's specific profile and pre-determined thematic categories (e.g. depression, social support, self-care, symptoms). Users can additionally carry out their filtering by theme, story type, age, gender, and community. Stories can also be hidden based on selected triggers. |
| Individual stories | Individual stories are headed up with brief information about the story type and duration. Story content is chunked into text-message-style bubbles for ease of reading. Users can choose to bookmark a story. Audio versions of stories are also provided. |
| Reactions | Users can respond to stories using a set of five 'reaction' buttons/tags: (i) hopeful, (ii) inspiring, (iii) helpful, (iv) relatable and (v) emotional. Users can select one reaction per text bubble. Reactions are saved and can be accessed by the user in the 'Saved' section of the website. |
| Story-based reflections | At the end of each story, users are prompted to answer a set of optional reflective questions, such as 'How connected to this story did you feel?' and 'If you were to try something you learned in this story, what would it be and why?' |
| Saved | The 'Saved' section allows users to access any text bubbles that have been marked with reactions, as well as containing bookmarked stories. |
| My Notes | The 'My Notes' feature is a dedicated space for journaling and making private notes. Users can additionally choose from a set of optional writing prompts. |
| Tell My Story | The 'Tell My Story' feature allows users to write their own story using a series of optional prompts. Users can upload images or audio files too. A section of information and tips on tips and considerations for sharing one's personal story is additionally provided. Users can save and come back to this section whenever they like. They can also download an offline copy of their story upon completion and have the option to submit this story for inclusion to the Baatcheet library. |
| Resources | This contains psychoeducational resources and information about mental health support services. It also includes a 'Frequently Asked Questions' (FAQs) section with question-and-answer style information about using the website. |
| Account profile | Users can access personal usage stats (e.g. the number of stories they have read or submitted) and edit their profile, language and data management preferences. |
| Feedback | This feature allows users to provide feedback on website use and any troubleshooting issues. |
| Peer support | Peer support is offered in-person or remotely through 'Baatcheet Buddies' across four weekly check-in meetings lasting up to 30 minutes each.<br>• Meeting 1: Orient the participant to the Baatcheet intervention and explain how to use the website.<br>• Meetings 2 and 3: Check progress, address any difficulties using the website and assist the participant with sharing their own story (if desired).<br>• Meeting 4: Review the participant's experience of the intervention and its potential impacts; provide information about external sources of mental health support, as needed. |

## Home

An external home page containing information about the intervention, how to use it and potential benefits of participation. It also provides testimonials of youth advisors and researchers involved.

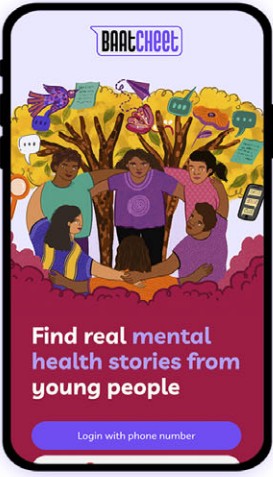

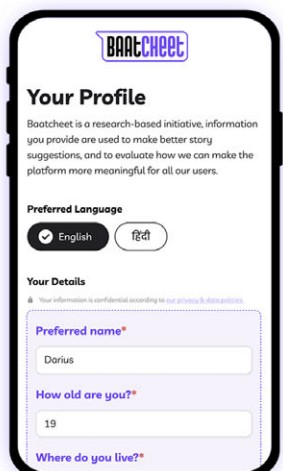

## Login

A login page providing the option to sign up using a mobile number or email address.

## Story Library

After logging in, users are taken to the story library page which is one of the key features of the website. This page contains stories which are arranged in sections including: stories randomly presented based on the user's profile; collections to select from; and stories featured by the Baatcheet team. Users can additionally filter by theme, story type, age, gender, community and hide stories based on selected triggers,

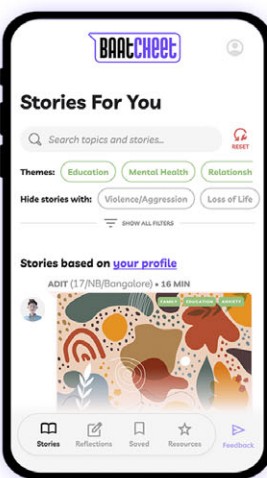

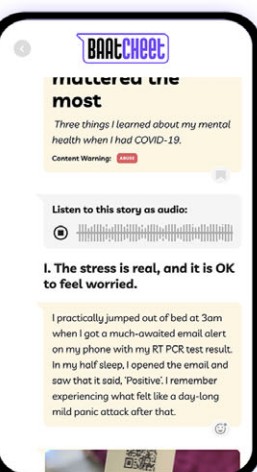

## Individual Stories

Individual stories contain brief information about the story type and duration. Story content is chunked into text-message style bubbles for ease of reading. Users can choose to bookmark a story. Audio versions of stories are also provided.

## Reactions

Each story text bubble allows users to "react" using a set of five predefined reactions identified through co-design activities. These reactions comprise: (i) hopeful; (ii) inspiring; (iii) helpful; (iv) relatable and (v) emotional. Users can select one reaction per text bubble. Reactions are saved and can be accessed by the user in the dedicated "Saved" section of the website.

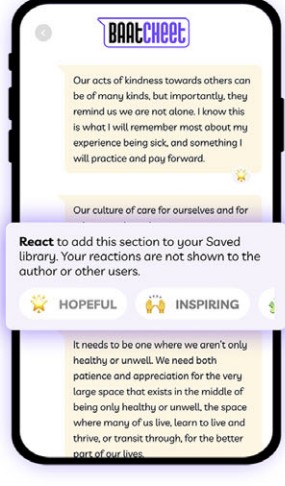

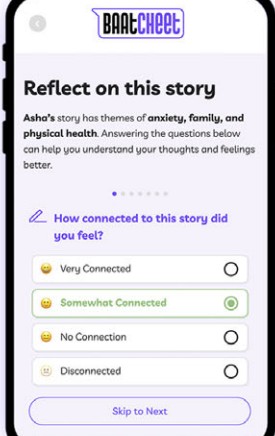

## Story Based Reflections

At the end of each story, users are prompted to answer a set of optional reflective questions.

**Figure 4.** Screenshots from the Baatcheet intervention website.

in its feature to contribute one's own story and the provision of brief peer support.

Studies of digital interventions consistently show that interventions that include an element of 'supportive accountability' (Mohr et al., 2011) are more effective and engaging than those that are fully automated or self-directed (Garrido et al., 2019; Grist et al., 2018; Grist et al., 2017; Hollis et al., 2017; Perret et al., 2023), with lower dropout rates (Clarke et al., 2015; Hollis et al., 2017). In this study,

## Saved

The Saved section allows users to access text bubbles from stories which they reacted to and to find their bookmarked stories.

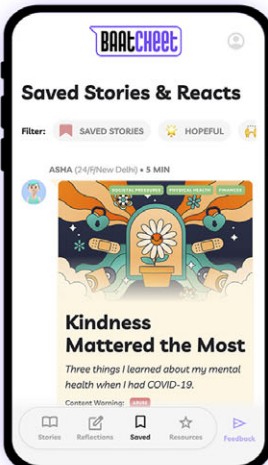

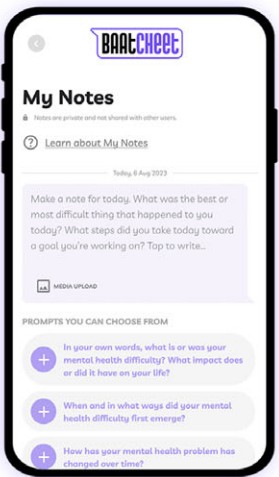

## My Notes

The My Notes feature is a dedicated space for journaling and making private notes. Users can additionally choose from a set of optional writing prompts.

## Tell My Story

The Tell My Story feature allows users to write their own story using a series of optional prompts. Users can upload images or audio files too. A section of information and tips is additionally provided. Users can save and come back to this section whenever they like. They can also download an offline copy of their story upon completion and have the option to submit this story for inclusion to the Baatcheet library.

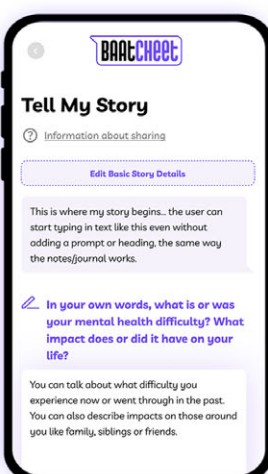

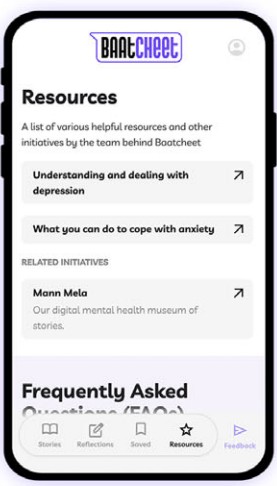

## Resources

The Resources section contains psychoeducational resources and information about mental health support services. It also includes a "Frequently Asked Questions (FAQs)" section with question and answer style information about using the website.

## Account Profile

The Account Profile allows users to access personal usage stats (e.g., the number of stories they have read or submitted) and to edit their profile, language and data management preferences.

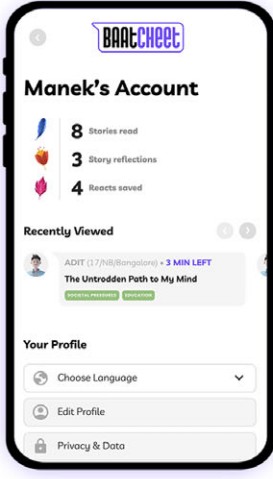

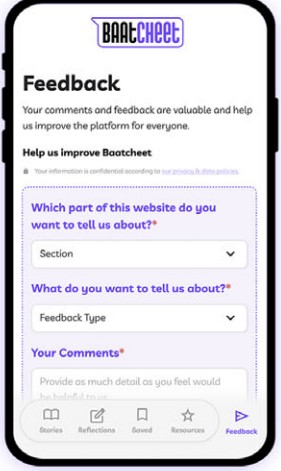

## Feedback

The Feedback feature allows users to provide feedback on website use and any troubleshooting issues.

**Figure 4.** Continued.

participants expressed preferences for motivational, relational and technical support from slightly older young people with lived/living experiences, aligned with other peer-supported interventions that employ slightly older youth (de Beer et al., 2022).

A key strength of our intervention development process was the involvement of young people at each stage, with prospective end-users shaping all aspects of the prototype in line with its original guiding principles. Confidence in the design decisions was

**Table 4.** User testing refinements to the Baatcheet website

| Feature | Revisions made to the intervention website |
|---|---|
| Home and about | • The title was revised to 'Find real mental health stories from young people'.<br>• Simplified and reduced the amount of text on the home page.<br>• Revised illustrations to make them more gender-neutral, inclusive and less abstract. |
| Login | • Revised the login process so that users can set content preferences when logging in for the first time. |
| Story library | • Filters and tags were revised to be more distinct.<br>• Included an option to filter stories based on the date of publication. |
| Individual stories | • Included additional animated videos to accompany written stories. |
| My notes | • Simplified language for story-based reflections.<br>• Redesigned the 'My Notes' and 'Tell My Story' sections to make them distinct. |
| Resources | • Included links to external psychoeducational resources previously co-developed with young people on understanding and coping with common mental health problems.<br>• Included local mental health support information and helpline numbers. |

strengthened by working with a YPAG and a relatively large group of 42 co-design participants that was balanced according to age, grade and gender. While this study took place in three types of low-resource urban settings, further work would be needed to confirm the generalisability of findings to other mental health problems, settings beyond those included in this study and with young people with differing levels of digital access.

Further development of the intervention could incorporate an expanded library of stories covering a wider range of problems in addition to depression and anxiety, enhanced personalisation and a concerted focus on supporting users to contribute their own stories. The utility of the intervention may also be explored for new audiences such as parents or carers (Ng et al., 2024). Finally, future work on Baatcheet should consider scalable opportunities for integration into existing youth services (e.g. as part of school or university peer support or counselling provision).

## Conclusions

To the best of our knowledge, this study is the first systematic attempt to co-design a digital mental health storytelling intervention with young people. Such interventions may hold promise in India and other low-resource contexts where there is an urgent need to provide non-stigmatising and resource-efficient interventions to improve young people's mental health. A subsequent study is assessing the acceptability, feasibility, and potential impacts of Baatcheet in a target population of 16–24-year-olds in schools, universities and community centres.

**Open peer review.** To view the open peer review materials for this article, please visit http://doi.org/10.1017/gmh.2024.148.

**Supplementary material.** The supplementary material for this article can be found at http://doi.org/10.1017/gmh.2024.148.

**Data availability statement.** The data that support the findings of this study are available from the corresponding author, PPG, upon reasonable request.

**Acknowledgements.** We gratefully acknowledge the contributions made by colleagues to intervention and research activities across the various phases of the project, especially our youth advisors: Aakarsha Jagga, Aathira Gopi, Akshat Shukla, Baishali Das, Deepak Mahawar, Kehkasha, Mehak Cheema and technical advisor Hitesh Sanwal. Thank you for sharing your incredibly valuable insights and time with us. We are also grateful to the study partners including Hemnani Public School, S. E. S. Baba Nebh Raj Senior Secondary School, University College of Medical Sciences (UCMS), Delhi University, and The YP Foundation. We extend utmost thanks to the students and stakeholders who participated in co-design activities without whom this study would not have been possible. Finally, this study would not be complete without the efforts of our website development team from Lattice Innovations, especially Soura Bhattacharya.

**Author contribution.** PPG and DM conceptualised and designed the study. PPG, DMittal, SA and AK led data collection, and analysis and contributed to the manuscript. PPG led the drafting and editing of the manuscript, which was critically revised by DM. All other authors read, reviewed, provided feedback and approved the final manuscript.

**Financial support.** This study was funded by a grant from Grand Challenges Canada (GCC) (Grant Number: R-GMH-POC-2210-55272) awarded to PPG and DM at Sangath, India and King's College London (KCL) UK.

**Competing interest.** None declared.

**Ethical statement.** Approval for the current study was obtained from the Institutional Review Board (IRB) of Sangath, the implementing institution in India (Reference number: PG_2022_84, dated 17.01.2023). The involvement of stakeholders in Stage 1 was consultative and therefore research consent was not obtained. For Stages 2 and 3, informed written assent (for youth aged under 18 years) or consent (for youth above 18 years) was obtained before participation, with parents/guardians providing additional consent for young people aged under 18 years. Detailed participant information sheets were provided, along with assent/consent forms in English and Hindi languages. Participants were provided with honorarium payments and certificates for their time and contributions. Participant welfare was prioritised throughout the study. Wherever possible, meetings were purposely scheduled at convenient times and in familiar locations. Activities were conducted bilingually (in Hindi and English) to maximise language accessibility. In addition, signposting to sources of mental health support, safeguarding and complaints procedures were shared with participants at the beginning of the co-design activities.

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
