## [Reviewer Report]

This paper summarizes the research and methods used to develop and evaluate Baatcheet, a peer-based online program for story sharing among young Indians with mental health challenges. The overall program shows great promise for an important priority. However, the paper was lacking in depth and breadth exactly how they achieved the varied steps. I would encourage authors to consider the following points in going forward.

1. Empowered telling of recovery stories among peers has arisen to become a major force for recovery-based systems. Unfortunately, this paper did not use lessons from previous similar efforts to guide their research questions and hypotheses, design, analyses, and findings. The paper notably cited Slade et al (2021) and NEON but then failed to detail what about NEON informed their decisions. The authors also note absence of this kind of work via LMIC but then failed to opine how LMIC might be relevant.

2. Let me be clear; I hugely endorse the co-design, community-based participatory research approach here. The problem is lack of detail so the reader cannot determine what was done nor establish validity of completed activities. There is much research and development on paradigms to adapt interventions for new settings (FRAME, Stirman et al., 2019) as well as address actual implementation of these efforts (Consolidated Framework for Implementation Research; Damschroder et al., 2022). The paper refers to MoSCoW as a possible method with no detail.

3. Qualitative methods are mentioned in several places without the detail readers need to understand the rigor of approaches.

4. One of the strengths of this paper is being grounded in India. But the paper reported two versions of Baatcheet (in Hindi and English). First, this suggests the overall process occurred independently from each other. Second, might not Hindi and English versions be different from each other?

5. Credit to the authors is noted for trying to somehow test impact of Baatcheet which seemed to incorrectly rest on use of SUS.

---

## [Reviewer Report]

Thank you for this really well written and easy to read manuscript.

I recommend it to be accepted as it is. There were font and paragraph issues on pages 2, 8 and 41. Please check.

---

## [Reviewer Report]

Thank you for the opportunity to review this manuscript, which provides a thorough description of the co-design development of Baatcheet. I would like to flag up a few points where further clarifications could be made:

1. p.9 line 16. Where the study design is described, with reference to Figure 1, it would be great to see a reflection of the involvement of the YPAG and Technical Advisory Group in relation to the three stages. The figure suggests that the YPAG and TAG were iteratively consulted between stages, but this is not clear from the manuscript narrative.

2. Alongside providing information on the involvement of Youth Advisors (p.13 line 14), it would be good to see a description of the composition and involvement of the Technical Advisory Group.

3. Table 1 – can it be specified what sessions are reported on in this table? I presume these are what is reported for Stage 2 FGDs (one per site) and co-design workshops (two for each participant), and Stage 3 user-testing sessions (up to two per group).

4. Does the participant N reported in Table 1 reflect the total number of participants involved in activities across multiple sessions in Stage 2 and 3? If that N is a composite, how many participants were involved per each co-design session? If the intention was to retain the same group throughout the sessions, was there drop out (and if so, were new participants introduced to boost numbers?)?

5. Regarding the ‘Story contents’ (p.21), is this more of a ‘wish list’ of preferred characteristics of stories, or is the plan to also use these as inclusion/screening criteria for stories selected for the website? Will there be a way to not feature stories that do not reflect these contents? Will story contents be fully derived in a bottom-up manner from this list of story characteristics, or will there also be a consideration of linkages with theory on what is know about key ‘active ingredients’ in personal narratives to e.g. create empathy and reduce stigma?

Minor points

6. p.13 line 22: it would be good to spell out that the 11 young people reflected 9 females, 2 males. Currently the meaning of F and M is implied.

7. p.14 line 46: does the NGO abbreviation need spelling out?

8. Please check table numbering in main text – currently the order on p.18 jumps from Table 1 to Table 4.

---

## [Editor Report]

Please would the authors attend to the reviewers' comments. We request that the authors also ensure that the manuscript has been edited by a first language English speaker prior to re-submission.

---

## [Reviewer Report]

The authors have provided full and clear responses to the comments raised in my previous review. I have no further questions or suggestions, and recommend this manuscript for publication.